# Evaluating the Impact of Minimized GnRH and PGF_2α_ Analogues-Loaded Chitosan Nanoparticles on Ovarian Activity and Fertility of Heat-Stressed Dairy Cows

**DOI:** 10.3390/pharmaceutics17020274

**Published:** 2025-02-18

**Authors:** Mohammed E. A. Omar, Eman M. Hassanein, Ahmed M. Shehabeldin, Ottó Szenci, Abdelghany A. El-Shereif

**Affiliations:** 1Animal Production Research Institute, Agricultural Research Centre, Ministry of Agriculture, Dokki, Giza 12619, Egypt; m.elshafie@arc.sci.eg (M.E.A.O.); ahmedshehabeldin85@gmail.com (A.M.S.); 2Animal and Fish Production Department, Faculty of Agriculture (El-Shatby), Alexandria University, Alexandria 21545, Egypt; em.mostafa@alexu.edu.eg (E.M.H.); abdelghany.awad@alexu.edu.eg (A.A.E.-S.); 3Department of Obstetrics and Farm Animal Medicine Clinic, University of Veterinary Medicine, István u. 2, H-1078 Budapest, Hungary

**Keywords:** reproductive performance, Ovsynch, GnRH, PGF_2α_, chitosan–TPP nanoparticles, synchronization, ovarian activity, dairy cows, heat stress

## Abstract

**Objectives**: This study aimed to evaluate the effectiveness of gonadotropin-releasing hormone-loaded chitosan–TPP nanoparticles (GnRH-CNPs) and prostaglandin F_2α_-loaded chitosan–TPP nanoparticles (PGF_2α_-CNPs) within the Ovsynch protocol for enhancing reproductive performance in heat-stressed dairy cows. **Methods**: Thirty-six cyclic purebred Friesian cows not detected in standing heat for more than 90 days postpartum were randomly allocated to three treatment groups. The control group (OVS, n = 12) followed the standard Ovsynch protocol with conventional doses. The ½ OVS group (n = 12) received 5 µg GnRH-CNPs on days 0 and 9, along with 250 µg PGF_2α_-CNPs on day 7. While the ¼ OVS group (n = 12) was administered 2.5 µg GnRH-CNPs on days 0 and 9, with 125 µg PGF_2α_-CNPs on day 7. Ovarian follicular dynamics and corpus luteum (CL) development were monitored on days 0, 4, 7, and 9 of the protocol. Serum progesterone (P_4_) concentrations were measured throughout the synchronization period and on days 15 and 30 post-AI. Pregnancy was diagnosed on day 30 post-AI. **Results**: The ¼ OVS protocol achieved a significantly greater follicular response (*p* < 0.05) than other protocols. On day 4, following the first GnRH administration, the OVS group exhibited a higher number of subordinate follicles (*p* < 0.05) and a greater diameter of the dominant follicles (DFs), whereas the ¼ OVS group showed a greater subordinate follicle diameter (*p* < 0.05) and a higher number of DFs. On day 9, after PGF_2α_ administration, the ¼ OVS group maintained an elevated number of subordinate follicles, while larger subordinate follicle diameters were observed in the ½ OVS and OVS groups. No significant differences in DF numbers and diameters were observed among groups. P_4_ concentrations remained similar across groups during treatments. Compared to control, a significantly higher value of P_4_ concentration (*p* < 0.05) was recorded on day 15 post-AI in the ½ OVS group and on day 30 post-AI in the ¼ OVS group. These findings correlated with a higher pregnancy rate in the ¼ OVS group (65%) compared to the ½ OVS and OVS groups (40% in each). **Conclusions**: Nanofabrication reduced GnRH and PGF_2α_ dosage by 50% and 75% without impairing ovarian response and pregnancy rates. The ¼ OVS protocol notably enhanced the ovarian activity and fertility, highlighting the use of GnRH-CNPs and PGF_2α_-CNPs as promising and practical approaches to enhance the fertility in dairy cattle under heat stress (HS).

## 1. Introduction

Optimizing reproductive performance is essential in livestock production, as the main reproductive goal is to achieve successful pregnancies promptly after the voluntary waiting period ends in lactating cows [1]. However, improving individual animal production has presented new challenges concerning estrous expression in high-producer dairy cows [2,3]. This challenge is associated with reducing circulating estradiol (E_2_) and progesterone (P_4_) concentrations, resulting from increased metabolic clearance of steroid hormone [4,5]. Additionally, the fertility of dairy cows is highly susceptible to environmental stressors, such as temperature and humidity [6,7]. Elevated temperature leads to hyperthermia and heat stress (HS), which impair fertility through various mechanisms [8].

Hyperthermia, defined as an increase in core body temperature exceeding 39.5 °C, has been associated with significant health and fertility challenges in dairy cows, depending on its severity and duration [9]. Heat stress adversely affects production performance [10] and disrupts reproductive processes [10]. In high-producing cows, postpartum HS intensifies the negative energy balance, leading to a reduction in the dominant follicle (DF) size and alteration in follicular profile, including changes in glucose, insulin growth factor-1 (IGF-1), non-esterified fatty acid (NEFA), urea, and total cholesterol [11]. Several studies have demonstrated that hyperthermia disrupts follicular dynamics, increasing the proportion of large follicles, double ovulations, premature emergence of the pre-ovulatory follicle, and elongated period of follicular dominance [12,13,14,15]. Furthermore, HS negatively affects estrous behavior, as pre-ovulatory follicles remain vulnerable to HS during the pre-ovulatory phase [16]. Elevated cortisol levels during HS suppress the GnRH response and pre-ovulatory luteinizing hormone (LH) surge, which is necessary for stimulating ovulation [17]. Plasma E_2_ levels are diminished under HS, compromising estrous expression [18]. Increased glucocorticoid levels further inhibit sexual behavior induced by E_2_. Additionally, the reduction in LH levels under HS conditions leads to the development of dominant follicles in a low-LH environment, subsequently reducing E_2_ secretion from these follicles and weakening estrous expression. Furthermore, low P_4_ levels, along with reduced E_2_, contribute to instances of silent heat in cows and heifers. As a result, HS leads to silent heat, diminishing the duration and intensity of estrus and ultimately reducing mounting activity [19,20].

The environmental conditions contributing to HS can be estimated from the temperature–humidity index (THI), calculated based on ambient temperature and relative humidity [21]. A THI below 68 is considered outside the thermal danger zone for dairy cows, while mild signs of HS appear at THI levels between 68 and 72. Reduced production and reproductive performances occur at THI 72 or higher [22]. Fertility can be modestly improved through several effective environmental management, such as providing shades, misters, evaporative cooling, fans in barns, overfeed bunks, and holding pens [21,23]. Additionally, the use of timed artificial insemination (TAI) protocols at first service eliminates the necessity of detecting estrus and allows for precise timing of AI relative to ovulation [24], representing another strategy to enhance fertility during HS [25].

TAI protocols implemented during HS conditions can enhance fertility by improving ovarian function [24]. This approach helps replace heat-stressed follicles with healthier ones that are more resistant to adverse environmental conditions and have a higher fertilizing capacity [26]. Moreover, TAI applications can effectively increase the conception rates, reduce days open, and lower the number of services pre-conception during the summer months [27]. A previous study indicated that synchronized cows with TAI protocols exhibited a higher pregnancy rate than non-synchronized females during elevated THI periods [25]. Similarly, several studies have implemented various TAI protocols, such as Ovsynch [28,29,30], Double Ovsynch [31], G-6-G [30], and PG-3-G [32], demonstrating their efficacy in improving fertility outcomes among cows experiencing HS.

The Ovsynch protocol, involving GnRH and PGF_2α_ administrations, is a widely recognized reproductive management strategy for synchronizing cattle ovulation [24]. However, conventional hormone administration within TAI protocols faces limitations, including high costs and a short half-life, especially GnRH, which lasts only 5–6 h in summer, leading to insufficient LH release [33,34]. Nanodrug delivery systems offer an innovative solution to enhance the efficacy of biological agents, including drugs and hormones. Unlike conventional formulations, nano vehicles-based drug formulations exhibit diverse pharmacokinetic properties [35]. Their tiny size, high surface area-to-mass ratio, and varied surface morphology and charge allow nanodrugs to interact differently within biological systems [36]. These unique characteristics can be optimized by carefully selecting specific carrier materials and controlling fabrication processes. This resulted in improved drug delivery to target sites while overcoming biological challenges such as the reduced biological half-life of hormones, decreased stability, and rapid degradation during transport in the bloodstream. Various encapsulation vehicles have been utilized for GnRH, including chitosan–dextran sulfate NPs [37], chitosan–TPP NPs [35,38,39], and chitosan alone [33]. Recent studies have demonstrated the effectiveness of nano-formulated GnRH across different nanocarriers in improving reproductive outcomes in farm animals, such as rabbits [38,40], goats [39,41], and buffalos [40]. These formulations have achieved similar or better outcomes while reducing the conventional hormone portion by 50% [37,39,40,41] to 75% [38]. Despite these promising findings, no studies have been obtained on dairy cattle.

Given these promising results, researchers are increasingly focusing on developing reduced hormonal doses that are as effective as conventional doses in regulating the estrous cycle while maintaining productivity [42,43]. This study, therefore, aims to evaluate the efficacy of GnRH conjugated to CNPs, along with PGF_2α_-CNPs, at half or quarter doses within a modified Ovsynch protocol compared to the conventional full dose of bare GnRH and PGF_2α_ dose in the standard Ovsynch protocol. The objective is to evaluate this novel method for enhancing the reproductive performance of heat-stressed cows in Egypt.

## 2. Materials and Methods

### 2.1. Hormones-Conjugated Chitosan Nanoparticles Fabrication and Evaluation

The ionic gelation process, as depicted by Gallab et al. [40], was applied to synthesize the bare chitosan and TPP nanoparticles (CNPs). Highly purified chitosan (Alpha Chemika^®^, Mumbai, India) with a deacetylation percent of over 85% and a low molecular weight, and sodium tripolyphosphate (TPP, Thermo Fisher^®^, GmbH, Karlsruhe, Germany) were utilized for polymeric nanocarrier formulation. Initially, bare CNPs were fabricated by gradually adding the aqueous solution of TPP (0.1 g/dL) into the aqueous acid solution of chitosan (0.1%, *w*/*v*) in a 1:2 ratio, respectively. Meanwhile, GnRH solution (Receptal^®^, as buserelin acetate, MSD, Intervet International GmbH, Unterschleißheim, Germany) was drop-wise added to the pre-structured CNPs in a 1:1 ratio to manufacture GnRH-CNPs. Similarly, PGF_2α_-CNPs were fabricated following the identical procedure incorporating a PGF_2α_ hormone solution (Estrumate^®^, cloprostenol, Vet Pharma, Friesoythe, GmbH, Friesoythe, Germany), albeit with a modified addition ratio of (2 CNPs:1 PGF_2α_). The mixtures were adjusted to a pH of 6.5 and then gently stirred for 60 min at 800 rpm. Subsequently, they were incubated overnight to ensure that the hormonal adsorption on the surfaces of the nanoparticles was achieved.

Concerning nanoparticle characteristics assessment, the loading efficiency (LE%) of CNPs’ hormones was measured using undercooling centrifugation at 1200× *g* for 20 min to separate the CNPs and hormones–CNPs solution from the aqueous medium. The supernatant of the CNPs solution was used as a blank to quantify the amount of free hormone in the supernatants of the PGF_2α_-CNPs and GnRH-CNPs solution using a UV spectrophotometer (Optizen pop, Mecasys Co., Ltd., Daejeon, Republic of Korea) at a wavelength of 280 nm [44]. Subsequently, the loading efficiency percentage was calculated using the formula [total hormone − free hormone in the supernatant/total hormone × 100]. Furthermore, Zetasizer (Malvern Instruments, Malvern, UK), depending on the dynamic light scattering (DLS) procedure, was used to measure the particle size, polydispersity index (PdI), and zeta potential for all fabricated nanoparticles. The samples were measured in triplicate, and each parameter’s average value (±SD) was estimated.

### 2.2. Ethical Approval

The study protocol received approval from Alexandria University’s Research Committee and was authorized by the Faculty of Agriculture under reference Alex. Agri. 082410327.

### 2.3. Animals and Management

The current study was conducted under field conditions at the Sakha Animal Production Research Station in Kafr El-Sheikh Governorate, Western, Egypt, 31°06′25.20″ N, 30°56′26.99″ E, which belongs to the Animal Production Research Institute (APRI), Agricultural Research Center (ARC), Ministry of Agriculture. It was conducted in collaboration with the Animal and Fish Production Department of the Faculty of Agriculture at Alexandria University.

The procedures and experimental protocols followed the guidelines outlined in the “Guide for the Care and Use of Agricultural Animals in Research and Teaching” [45]. The study involved clinically healthy heat-stressed Purebred Friesian cows weighing 450 and 550 kg, aged 3.5 to 5.5 years, with 1 to 3 parities, and a body condition score ranging from 2.75 to 3.0 on a five-point scale [46]. Cows were milked twice daily, averaging a milk yield of 18.0 ± 5.0 kg/head/day. They were housed in a semi-open, shaded yard. They were fed a balanced diet to meet their maintenance and production requirements. During the summer season, cows received a diet consisting of a concentrate feed mixture, berseem hay, corn silage, and rice straw to fulfill their nutritional requirements, following the National Research Council guideline [47], along with free access to fresh water.

### 2.4. Collection of Meteorological Data

Meteorological data, including air temperature (T_a_, °C) and relative humidity (RH, %) for the study location in Kafr El-Sheikh governorate, were obtained from the Central Laboratory for Agricultural Climate at Agricultural Research Center, Ministry of Agriculture and Land Reclamation, Egypt. The data were sourced from the meteorological station within the governorate and covered the study period from early July to late August. Daily minimum, maximum, and average were derived from hourly temperature and humidity measures. Utilizing this, the daily minimum, maximum, and average THI values for the experimental months were calculated using the following equation [6]:$$THI=\left(0.8\times T_{a}\right)+\left(\frac{RH}{100}\times \left(T_{a}-14.4\right)\right)+46.4$$

### 2.5. Experimental Design

Thirty-six cyclic Purebred Friesian cows not detected in standing heat for more than 90 days postpartum were randomly assigned to three groups, each receiving a different Ovsynch protocol, as depicted in Figure 1. The first group (n = 12; control group) was treated with the standard Ovsynch protocol (OVS), consisting of two intramuscular (i.m.) injections of 10 μg GnRH (Receptal^®^) on days 0 and 9, and 500 μg i.m. injection of PGF_2α_ (Estrumate^®^) on day 7. In the second group (n = 12), a modified Ovsynch protocol (½ OVS) was applied, using a reduced dose of GnRH and PGF_2α_. Cows received 5 μg GnRH-CNPs on days 0 and 9 and a 250 μg injection of PGF_2α_-CNPs on day 7. The third group (n = 12) followed another modified protocol (¼ OVS) with a reduced dose of GnRH and PGF_2α_, receiving 2.5 μg GnRH-CNPs on days 0 and 9 and a 125 μg injection of PGF_2α_-CNPs on days 7. All cows were artificially inseminated 16–20 h following the second GnRH injection, using high-quality frozen–thawed semen (30 × 10^6^ sperm/0.5 mL) from a highly fertile bull. All cows were subjected to the ovulation synchronization protocol simultaneously, ensuring that they experienced identical environmental conditions throughout the experiment.

### 2.6. Hormonal Analysis

Blood samples were withdrawn in the early morning, at 8 a.m., before feeding (10 mL each) on days 0, 4, 7, and 9 of each synchronization protocol and on days 15 and 30 post-AI. The serum was separated by centrifuging the blood samples at 700× *g* for 20 min and then stored at −20 °C for later hormonal analysis. Serum P_4_ concentrations were determined using enzyme-linked immunosorbent assay (ELISA) kits (Monobind Inc., Lake Forest, CA, USA), with a detection limit of 0.11 ng/mL for P_4_. The intra-assay and inter-assay coefficients of variation were 9.3% and 9.9%, respectively.

Additionally, the area under curve (AUC) values for P_4_ concentrations were calculated to monitor the immediate response of treatment for the experimental groups (OVS, ½ OVS, and ¼ OVS) across specific time intervals. These intervals were defined as follows: D4 (∆ AUC between days 0 and 4), D7 (∆ AUC between days 4 and 7), and D9 (∆ AUC between days 7 and 9). The calculations were performed using the trapezoidal rule, which combines data points over time to provide a cumulative measure. AUC values were calculated for each group across the mentioned intervals, applying the following equation [48]:$$AUC=\sum_{i=1}^{n-1}\frac{\left(y_{i}+y_{i+1}\right)}{2}\times (x_{i+1}+x_{i})$$
where y_i_ and y_i+1_ indicate the P_4_ concentrations at two sequential time points; and x_i_ and x_i+1_ denote the corresponding time points on days 0, 4, 7, and 9.

### 2.7. Ultrasound Examination

The ovarian structures were examined using transrectal ultrasonography, according to the protocol described by Fricke [49]. A real-time B-mode veterinary ultrasound device (Esaote Pie Medical Aquila Pro Vet with 6.0/8.0 MHz LA Rectal Veterinary, Esaote, Italy) was used. The ovarian structures were scanned on days 0 (before the first GnRH administration to confirm ovarian status), 4 (to determine the response to the first GnRH injection), 7 (before the PGF_2α_ administration), and 9 (just before the second GnRH administration). The total number of follicles, the diameter of subordinate follicles (≥0.3 to <1.0 cm), and the dominant follicles (DFs, >1.0 cm) were recorded, along with the count of corpora lutea (CLs). Additionally, pregnancy diagnosis was conducted on day 30 post-AI by examining the uterine contents. The pregnancy rate (PR) was calculated as follows: PR = [number of pregnant cows on day 30/total number of serviced cows × 100].

### 2.8. Statistical Analysis

The normality of the data was assessed using the Shapiro–Wilk (W) test from the UNIVARIATE procedure of SAS (SAS Institute Version 9.0. SAS Institute, Inc., Cary, NC, USA) [50]. Follicular dynamics, serum P_4_ concentration, and P_4_ area AUC data were analyzed using the GLM of SAS [50] to assess significant differences among Ovsynch protocols and across various sampling days within each protocol. The minimum, maximum, average temperature, and the THI values were calculated using Microsoft Excel (Microsoft Corporation, Redmond, WA, USA). All statistical graphs were generated using GraphPad software (GraphPad Prism 9, San Diego, CA, USA). All results were presented as (means ± SEM). Mean comparisons were conducted using Duncan’s multiple-range test [51]. Additionally, pregnancy rate data were analyzed using the chi-square test, with statistical significance set at *p* < 0.05.

## 3. Results

### 3.1. Characterization of Chitosan and GnRH-Loaded Chitosan Nanoparticles

As illustrated in Figure 2, the physicochemical properties of the unconjugated CNPs, including the average particle size, PdI, and zeta potential, were 74.8 ± 25.4 nm, 0.67, and +0.34 ± 0.4 mV, respectively. For the GnRH-CNPs, the average particle size was 94.4 ± 23.9 nm, with a PdI of 0.40 and a zeta potential of +12.0 ± 2.1 mV. The corresponding values for the PGF_2α_-CNPs were 149.5 ± 37.4 nm in size, a PdI of 0.548, and a zeta potential of +18.0 ± 6.04 mV. The hormone LE was also 90.5% for GnRH and 85.7% for PGF_2α_.

### 3.2. Environmental Conditions During the Study

During the study period, daily ambient temperatures were analyzed based on hourly measurements obtained from the meteorological station near the study location. The maximum temperature recorded was 35 °C, the minimum was 23 °C, and the overall average temperature was 28.1 °C. Relative humidity levels varied, with a maximum of 90%, a minimum of 62%, and an overall average of 67.3%. These environmental conditions resulted in THI values, as shown in Figure 3, ranging from a minimum of 68.7 to a maximum of 91.2, with an overall average of 78.3 ± 5.2 for July and August. This suggests the presence of HS throughout the study period, as throughout June and July, most average THI values exceeded the threshold of 72.

### 3.3. The Ovarian Structure

The results of ovarian structure ultrasonography illustrated significant changes influenced by protocol-day progression (Table 1 and Figure 4). On day 0, all cows exhibited an ovarian activity without being detected in standing heat for more than 90 days postpartum. However, no significant differences were observed among groups for any ovarian structure parameters, including the total number of follicles (*p* = 0.15), the number and diameter of subordinate follicles (*p* = 0.24 and *p* = 0.76, respectively), and the diameter of DF (*p* = 0.20). However, regarding the number of DF, both OVS and ¼ OVS exhibited a tendency for higher follicle numbers (0.33 ± 0.14 cm each, *p* = 0.07) compared to ½ OVS, which had no DF. These findings are presented in Table 1 and Figure 4.

On day 4, following the first GnRH administration, the total number of ovarian follicles and the number of subordinate follicles demonstrated a comparable trend across all groups during the various days of the protocol. A significant increase was observed in the OVS group, while the ½ OVS and ¼ OVS remained nearly stable compared to day 0 (Figure 4A,B). Consequently, the OVS group exhibited the highest number (*p* < 0.05) of total follicles (5.83 ± 0.37) on day 4, which was significantly higher than ½ OVS (3.50 ± 0.52), while ¼ OVS recorded an intermediate mean count of 4.50 ± 0.52. Furthermore, the OVS group recorded a superior number (*p* < 0.001) of subordinate follicles (5.50 ± 0.29) compared to that of both the ½ OVS and ¼ OVS groups (3.17 ± 0.44 and 3.50 ± 0.45, respectively). Regarding the subordinate follicle diameter, presented in Figure 4C, their diameter significantly increased in the ¼ OVS group, measuring 0.77 ± 0.07 cm. In contrast, the OVS and ½ OVS had diameters of 0.60 ± 0.02 cm and 0.61 ± 0.05 cm, respectively. However, it remained unchanged across all groups compared to day 0. In terms of DFs, the number of DFs on day 4 remained stable in the OVS and ½ OVS groups (0.33 ± 0.14 each), although there was a slight numerical increase observed in ½ OVS group, as shown in Figure 4D. In contrast, the ¼ OVS group showed a significant improvement (*p* < 0.05) in the number of the DFs (1.00 ± 0.17), attributed to a more considerable proportion of cows showing follicular response following the first GnRH administration. Notably, when comparing the treatment effects, the ¼ OVS group showed a significantly higher number of DFs than the other groups. A contrasting trend was observed in the DF diameter. The diameter of DF significantly increased in both the OVS and ¼ OVS groups. In contrast, the ½ OVS group experienced a non-significant decrease on day 4 (*p* > 0.05) compared to day 0 (Figure 4E). Notably, the OVS group recorded the largest DF diameter of 1.49 ± 0.13 cm, which was significantly higher (*p* < 0.05) than the diameter recorded in the other treatment groups.

As the protocol advanced, the total number of follicles and subordinate follicles significantly declined in the OVS group by day 7 compared to day 4 (Table 1 and Figure 4A,B). Conversely, these numbers remained stable in both the ½ OVS and ¼ OVS groups. However, the treatment groups (½ OVS and ¼ OVS) showed a greater total number of follicles, with values of 3.50 ± 0.67 and 4.83 ± 0.41, respectively, compared to the OVS group, which exhibited only 2.00 ± 0.35. Additionally, the ¼ OVS group exhibited a sustained increase in subordinate follicle numbers, reaching 4.50 ± 0.51 (*p* < 0.05), compared to the other groups (OVS: 2.00 ± 0.35 and ½ OVS: 3.00 ± 0.65). The diameter of subordinate follicles increased (*p* < 0.05) in the ½ OVS group on day 7 compared to day 4; however, there were no significant differences (*p* > 0.05) in follicle diameter among the group on this day (Figure 4C). The number of DFs declined significantly (*p* < 0.05) in the ¼ OVS group but increased notably (*p* < 0.05) in the ½ OVS group, with the OVS group showing complete disappearance of DF (Figure 4D). The diameter of DF remained unchanged in the ¼ OVS group, recorded at 0.33 ± 0.14 cm, while a slight but non-significant decrease (*p* > 0.05) was observed in the ½ OVS group, measured at 0.50 ± 0.15 cm (Figure 4E).

By day 9, the OVS group showed a notable increase in the total number of follicles and subordinate follicles, as shown in Table 1, suggesting the emergence of new follicles following the administration of PGF_2α_ (Figure 4A,B). This increase was statistically comparable to that of the ½ OVS group (4.50 ± 0.45 for OVS vs. 4.17 ± 0.44 for ½ OVS group). However, it was significantly lower (*p* < 0.05) than the number recorded in the ¼ OVS group (6.17 ± 0.41). The ¼ OVS group displayed the highest number of follicles (5.33 ± 0.38), highlighting its superior effectiveness compared to the OVS and ½ OVS groups, which had 3.50 ± 0.34 and 3.17 ± 0.44, respectively. Concerning the diameter of subordinate follicles, it increased significantly, with the OVS and ½ OVS groups both measuring 0.88 ± 0.07 cm, which was significantly higher than that of the ¼ OVS group, which measured 0.66 ± 0.02 cm (Figure 4C). In addition, both the number and diameter of DF increased in all groups, with no significant differences (*p* > 0.05) observed among them (Figure 4D,E).

Concerning the luteal activity, the number of corpora lutea (CLs) was significantly influenced by the days of experimental protocols, as illustrated in Table 1 and Figure 4F. On day 0, the OVS group showed the highest number of CLs (1.00 ± 0.00), while the ½ OVS and ¼ OVS groups exhibited 0.83 ± 0.11 and 0.67 ± 0.14, respectively. However, the differences were not statistically significant (*p* = 0.09). By day 4, the number of CLs remained relatively stable across groups, with no significant differences (*p* = 0.08). However, in the ½ OVS group, the number of CLs numerically decreased by day 4, as fewer cows exhibited CLs compared to day 0 (Table 1 and Figure 4F). The CL numbers were consistent across all groups (1.00 ± 0.00) on day 7. Conversely, on day 9, all groups had a dramatic reduction in CL number (*p* < 0.05) following the PGF_2α_ administration. Despite these changes, the different protocols did not significantly affect CL numbers at any specific time (Figure 4F).

### 3.4. Serum P_4_ Profile

Serum P_4_ concentrations throughout the study also demonstrated significant variations (*p* < 0.05), closely associated with the recorded number of CLs during the synchronization protocol days (Figure 5A). On day 0, there were no significant differences in P_4_ concentrations across all groups. However, by day 4 (day of the first GnRH administration), P_4_ concentrations decreased significantly (*p* < 0.05) compared to day 0. This decline is associated with the reduction in the number of cows exhibiting CLs, suggesting as a response to the natural ovarian activity, likely influenced by variation in the stage of ovarian follicles at the initiation of the protocol.

In contrast, on day 7, P_4_ concentrations increased, as all cows presented at least one functional CL (>1 ng of P_4_); however, these differences were not statistically significant (*p* > 0.05) across the groups. The lowest P_4_ concentrations (*p* < 0.05) were observed on day 9 in response to PGF_2α_, whereas the highest concentrations (*p* < 0.05) were recorded on days 15 and 30 post-AI. The ½ OVS protocol exhibited a pronounced luteotropic impact, characterized by a significantly higher P_4_ concentration on days 9 and 15 post-AI (*p* < 0.05) compared to the control group. The ¼ OVS protocol, on the other hand, recorded the highest P_4_ concentration on day 30 post-AI, comparable to the ½ OVS group (*p* < 0.05).

Moreover, the observed alterations in the area under the curve (AUC) of serum P_4_ (Figure 5B) serve to confirm these findings, providing a comprehensive measure of cumulative P_4_ concentrations across defined time intervals. The AUC values reflect dynamic hormonal variations during the application of various synchronization protocols and are associated with the recorded number of CL observed during the protocol days.

During the initial intervals (days 4 and 7), the P_4_ AUC values were observed to be low and did not exhibit significant differences across the treatment groups (*p* = 0.7 and *p* = 0.9, respectively). A notable reduction in P_4_ AUC was identified on day 9, which corresponded with the luteolytic effects of PGF_2α_ administration. After PGF_2α_ administration, cows in the ½ OVS group exhibited a higher P_4_ AUC value (3.3 ng/mL·d, *p* = 0.04) compared to those in the ¼ OVS and OVS protocols, which recorded P_4_ AUC values of 2.7 and 2.5 ng/mL·d, respectively. These findings highlight the temporal effect of differing synchronization protocols on serum P_4_ dynamics.

### 3.5. Pregnancy Rate

None of the experimental cows showed estrous signs following the first injection of GnRH or PGF_2α_. All animals were fixed-time inseminated after the second injection of GnRH. Cows treated with the ¼ OVS protocol had the highest pregnancy rate (8/12; 60%), which was significantly higher (*p* < 0.05) compared to the ½ OVS and OVS groups (5/12; 40% for each). Meanwhile, there were no substantial differences between ½ OVS and OVS groups.

## 4. Discussion

This study aimed to evaluate the efficacy of administering reduced doses of GnRH-CNPs and PGF_2α_-CNPs within the modified Ovsynch protocol—using half or quarter doses (½ OVS and ¼ OVS, respectively)—compared to the standard OVS protocol. The study assessed their impact on ovarian structures, P_4_ concentrations, and fertility outcomes in dairy cows exposed to HS during the summer season in Egypt.

Nanoparticle size is crucial for crossing mucosal barriers and achieving intracellular uptake [52]. Most drug delivery nanoparticles range from 50 to 250 nm, a size that facilitates efficient passage through biological barriers and enhances cellular uptake [53,54]. In our study, the increase in the particle size of CNPs following hormone loading agrees with earlier findings [39,40,55], suggesting the presence of molecular interactions and aggregations between the hormone and the nanocarrier.

The surface charge of nanoparticles, expressed as zeta potential, is crucial for stability, with high values (whether positive or negative) indicating strong particle stability due to repulsion that prevents aggregation [54,56]. Positive zeta potential, in particular, can enhance cellular uptake, which is essential for cell viability [53,57]. In this study, GnRH-CNPs and PGF_2α_-CNPs exhibited positive zeta potentials, indicating high stability and potential for efficient cellular uptake. In addition, incorporating chitosan as a nanocarrier significantly enhances these characteristics, promoting improved mucosal absorption and providing greater resistance to enzymatic degradation [58].

LE is a vital factor influencing the biological activity of nanoconjugates, as higher LE helps protect peptide hormones from enzymatic degradation, thereby extending their half-life. The chitosan–TPP nanoparticle system achieved 90.5% LE for GnRH-CNPs and 85.7% LE for PGF_2α_-CNPs in this study. These values are supported by previous reports [38,39,40,55], which showed high LE values for various GnRH analogs conjugated with CNPs, ranging from 87.7% to 91.2%. This supports the efficacy of the chitosan–TPP system in maintaining hormone integrity and enhancing its biological activity [54]. In contrast, Rather et al. [35] reported a lower LE of 69% for GnRH, further highlighting the effectiveness of this study’s preparation and conjugation methods.

Previous studies have identified a THI of 72 as the critical threshold based on the observed relationship between mean or maximum THI on AI day and subsequent pregnancy rates [25,59]. During the study period, dairy cows were exposed to HS from early July to late August, with an overall average THI of 78.3 ± 5.2. These values consistently exceeded the critical threshold of 72, indicating significant severe HS exposure, adversely affecting productive and reproductive performance [60]. Heat stress (HS) results in a deviation of body temperature from its resting state and disruption of its physiological and productive health [61]. The increased body temperature (hyperthermia) associated with HS adversely affects reproductive hormones [10]. The reduction in conception rates during elevated ambient temperatures can range between 20% and 30%, as opposed to the winter season [62]. It is well documented that HS occurring before AI significantly affects conception rates [63]. This stress decreases the duration and intensity of behavioral estrus, leading to a lower proportion of cows being identified in estrus under HS conditions [15].

This study employed the Ovsynch and modified Ovsynch protocols as TAI methods to synchronize ovulation in heat-stressed cows not exhibiting estrous signs for more than 90 days postpartum. The synchronization of ovulation causes the cessation of the existing wave of follicular development, followed by the initiation of a newly synchronized wave. This cessation of the existing follicular wave was observed in the control and the ¼ OVS groups following the administration of PGF_2α_ on day 7. However, the ½ OVS group exhibited a different response, described by the continuous stimulation and development of the follicular wave from day 0, ultimately reaching the preovulatory stage by day 9. This variation can be attributed to the differences in the developmental stages and response of the ovarian follicle population at the initiation of the protocols. The induced interruption of ovarian follicular development can be achieved by administering GnRH. For the GnRH-based protocol, the administration of the hormone results in either ovulation or regression of the DF, depending on its stage of development [61,64]. The Ovsynch protocol is a widely applied regimen for inducing estrus, significantly improving the reproductive efficiency of dairy cattle [65]. As a TAI protocol, it provides numerous benefits for facilitating early postpartum reproductive activity. It minimizes the necessity for detecting estrus and conserves labor by enabling scheduled times for AI. Additionally, timed AI may prove especially effective during hot weather conditions, as HS can complicate estrous detection [28,29,30]. These improvements may be attributed to increased estrous expression and ovulation of anovular cows, as well as elevated P_4_ levels during the selection of DF [64,66,67]. These findings highlight the potential benefits of TAI protocols in improving reproductive outcomes under challenging subtropical environmental conditions.

Recent advances in nanodrug manufacturing have demonstrated that transforming hormones into nano-forms can enhance their biological activity by modifying their physicochemical properties [52,68]. Our results unveiled the potential of the ¼ OVS protocol, which was the most effective in reducing follicle numbers by day 4 after the first GnRH injection. At the same time, a similar effect was observed with OVS on day 7. The ½ OVS protocol did not significantly influence follicle numbers on any sampling day. These findings indicated that ¼ OVS hormones could maintain adequate bioactivity to effectively stimulate follicular turnover, likely due to increased LH secretion from sustained hormone release. This is consistent with findings in goats, where full or half doses of GnRH-CNPs did not significantly impact follicle numbers [41]. It is well established that the prominent physiological roles of GnRH analogs are to stimulate folliculogenesis and induce ovulation. The main goal of the first GnRH injection in Ovsynch is to induce ovulation or luteinization of developing follicles, reducing follicle numbers and facilitating the emergence of a new follicular wave [41].

Our findings also revealed that follicular diameter was the largest in the ¼ OVS group on day 4 and the ½ OVS group on day 7 following the first GnRH injection. These observed differences can be associated with the variations in the implemented protocol, as no significant differences were detected in the number and diameter of subordinate follicles among groups before the initiation of protocols. Previous studies suggest that nano-GnRH enhances follicular growth more effectively than conventional GnRH [40]. Nano-protection of GnRH is believed to promote the replacement of healthy follicles with larger ones, leading to higher P_4_ concentrations than conventional GnRH. This is because nano-GnRH extends the hormone’s half-life in the bloodstream to 10 h, nearly double that of traditional GnRH [33,37,39,69].

In the ¼ OVS protocol, the number of DF was significantly higher on day 4 but decreased by day 7, while the OVS and ½ OVS protocols showed similar numbers of large follicles throughout. Comparable results were observed in goats treated with either bare or half doses of GnRH-CNPs, demonstrating a significant reduction in the number of large follicles compared to conventional GnRH [41].

In large animals, multiple and sequential GnRH doses are typically required to achieve follicle diameters of around 10 mm [70]. In this study, the DF diameters peaked in the OVS group on day 4 and in the ½ OVS, indicating a more robust follicular response to the first GnRH administration. In buffaloes treated with the Ovsynch protocol, the diameter of the DF was reported to be 8.1 mm [71] and 9.7 mm [33], indicating that GnRH alone may not elicit an optimal follicular growth response. Similarly, in our study, the diameter of the DF remained below 10 mm across all protocols on days 4 and 9. Raval et al. [70] also found similar results in anestrous buffaloes treated with double doses of the Ovsynch protocol, where the DF reached 10.7 mm. Additionally, it has been reported that buffaloes treated with a half dose of GnRH-CNPs had higher E_2_ concentrations on day 10 (estrus) compared to the control Ovsynch group, likely due to the presence of larger follicles with greater diameters in the nanoparticle-treated group [40]. In the current study, these variations in follicular response may be attributed to differences in ovarian activity and follicular wave patterns at the start of the protocols. Furthermore, the physiological impact of HS, which is known to impair ovarian function [15,16], may also contribute to the observed variability.

The current study shows that the number of CLs remained consistent on days 0 and 4 within the OVS and ¼ OVS groups; however, within the ½ OVS group, the CL count decreased numerically by day 4 following the first GnRH injection. This decrease can be attributed to differences in variations in ovarian structure among groups at the initiation of the synchronization protocols. In the ½ OVS group, the number of cows exhibiting CLs was reduced on day 4 compared to day 0 due to the natural dynamics of the ovarian cycle and the developmental stages of the CL at the initiation of the protocol. In contrast, the control and ¼ OVS displayed no change in the number of cows with CLs between days 0 and 4. This reduction in CLs within the ½ OVS group also corresponded with a decline in P_4_ levels. This result highlights a potential association between this group’s ovarian structures and hormonal fluctuations. By day 7, all protocols exhibited one functional CL per cow, demonstrating the effectiveness of the first GnRH injection—whether in conventional form or as a half or quarter dose of GnRH-CNPs—in inducing the DF, triggering ovulation, and facilitating subsequent CL formation. According to the Ovsynch protocol, administering GnRH during the early luteal phase may enhance CL functionality by promoting LH release from the anterior pituitary, which leads to the luteinization of granulosa and thecal cells [24,72,73]. The similar response to the first GnRH injection across different dosages in all protocols suggests that GnRH-CNPs may enhance the biological efficacy of conventional GnRH by supporting a prolonged gonadotropin surge.

Moreover, all protocols showed similar outcomes following PGF_2α_ administration, as complete luteal regression occurred by day 9, regardless of whether a whole, half, or quarter dose of PGF_2α_-CNPs or conventional PGF_2α_ was used. The CL is a well-known target for PGF_2α_, but a fundamental challenge to its activity is the rapid degradation of PGF_2α_ in the circulatory system [42]. Supporting these findings, Hashem et al. [41] demonstrated that reducing the PGF_2α_ dose by half did not compromise its luteolytic activity. Other studies have emphasized modifying PGF_2α_ analogs to ensure sustained hormone release over time [73]. Our results confirm that halving or quartering the PGF_2α_ dose as PGF_2α_-CNPs produced the same luteolytic effect as a total dose of conventional PGF_2α_. Similarly, GnRH-CNPs demonstrated intense epithelial penetration, promoting follicular growth and forming functional CLs [35,74] as more granulosa cells became available for luteinization [75].

In dairy cows, P_4_ plays a critical role in maintaining pregnancy at the beginning of gestation, with the CL serving as the primary source of P_4_ [76]. In this study, serum P_4_ concentrations consistently responded across all protocols following the first GnRH administration. However, after PGF_2α_ administration, cows in the ½ OVS protocol exhibited higher P_4_ concentrations than those in the ¼ OVS and OVS protocols. This enhanced P_4_ response with the half dose of GnRH-CNPs compared to the quarter dose may be linked to CL diameter. Similarly, Hashem et al. [41] reported that higher doses of GnRH-CNPs extended the luteal phase in synchronized estrous cycles of goats, evidenced by larger CL diameters and increased serum P_4_ concentration. In our study, serum P_4_ concentrations across all protocols from day 0 to day 9 were associated with the mean number of CLs. The significant variation in P_4_ concentrations between protocols on day 9 may be attributable to differences in CL diameter, which was more critical in the ½ OVS protocol than the ¼ OVS and OVS protocols. Elevated P_4_ concentrations on the day of PGF_2α_ administration may enhance the growth rate of the dominant follicles during the follicular phase [77]. Furthermore, modifying the Ovsynch protocol with GnRH-CNPs increased P_4_ concentrations on day 7, contributing to the larger follicle diameters observed [66]. In addition, the ½ OVS and ¼ OVS protocols can positively affect serum P_4_ concentrations during early pregnancy, specifically on days 15 and 30 post-AI. Enhanced follicular growth was linked to increased CL diameters, resulting in higher P_4_ concentrations on days 15 and 30 post-AI. This optimal luteal function promotes uterine receptivity and embryo survival [78], ultimately leading to improved fertility outcomes, such as a 75% conception rate [40]. The increase in CL diameter is directly related to the size of pre-ovulatory follicles and the number of granulosa cells, which are essential for forming luteal tissue [33,79].

Regarding pregnancy rates (PRs), our findings demonstrated that using GnRH-CNPs in the ½ OVS protocol had a comparable effect on PR to the conventional OVS protocol. However, the ¼ OVS protocol with GnRH-CNPs resulted in a substantial increase in PR, approximately 25% higher than the other protocols. Initially, we anticipated higher PRs in the ½ OVS and ¼ OVS protocols compared to OVS due to significant differences in P_4_ concentrations on day 15 across protocols. The lower PR observed in cows under the ½ OVS protocol, compared to the ¼ OVS protocol, may be due to early embryonic mortality resulting from HS. This is suggested by the modest increase in P_4_ levels recorded on day 30 post-AI. In cows subjected to HS, the intrauterine environment is adversely affected, resulting in diminished blood flow to the uterus and an increase in uterine temperature. These conditions impede embryonic development, heighten the risk of early embryonic loss, and reduce the success rate of inseminations [80,81]. In contrast, a more pronounced elevation in P_4_ levels was noted in the ¼ OVS and OVS protocols between days 15 and 30. These differences in P_4_ levels may be associated with variations in the diameter of the CL [40]. The increased bioavailability of GnRH leads to a prolonged LH pulse frequency in ovarian tissues, promoting follicle growth and improving oocyte quality [33,35], compared to conventional GnRH. CNPs also provide sustained release, high delivery efficiency, and an increased half-life of GnRH in the bloodstream [37]. It has been reported that buffaloes injected with half dose GnRH-CNPs had a higher conception rate (75% vs. 40%) and PR (60% vs. 20%) than those treated with conventional GnRH [40], likely due to increased ovarian activity and larger follicle and CL diameters. Recent studies have indicated that elevated concentrations of P_4_ on the day of PGF_2α_ administration can significantly enhance PR [40]. Furthermore, the correlation between serum P_4_ levels exceeding 1 ng/mL at the time of PGF_2α_ administration improved PR in buffaloes [77].

Despite the promising preliminary results of the study, notable limitations should be addressed. The study did not extend its evaluation to include pregnancy rates after 60–70 days, and it did not assess the long-term environmental and health impacts of prolonged use of nanoparticle-based hormones. While reduced doses demonstrate clear advantages, the specific mechanisms of action and the potential effects of nanoparticle accumulation in animal tissues require further investigation. Ultimately, the scalability and practical application of these nano-based protocols across diverse breeds, physiological status, management systems, and environmental conditions remain critical for future research.

## 5. Conclusions

This study’s findings reveal the potential of nanofabricated hormonal treatments in revolutionizing reproductive management in heat-stressed dairy cows. Nanofabrication of GnRH and PGF_2α_ enabled a 50% and 75% reduction in hormonal doses without compromising fertility outcomes compared to the full conventional doses. Among the protocols, the ¼ OVS protocol showed a more robust follicular response and higher pregnancy rates than the traditional OVS protocol, indicating that GnRH-CNPs can improve ovarian activity and P_4_ concentrations during early pregnancy. These findings support using GnRH-CNPs and PGF_2α_-CNPs as alternatives to conventional hormonal treatments, enhancing efficiency and reproductive success in heat-stressed dairy cows. Additionally, there may be potential economic benefits, making these protocols economically attractive for large-scale implementation in the future.

These preliminary findings have critical practical implications for the dairy industry. The ability to achieve similar or better reproductive outcomes with reduced hormonal doses translates into cost savings and reduced drug usage, aligning with efforts to minimize the adverse environmental impact of livestock production. Additionally, the nano-based protocols offer an effective solution during the low-breeding season, particularly under challenging conditions such as HS, where traditional approaches often fall short. By improving reproductive efficiency, these protocols contribute to maintaining herd productivity and profitability, especially in climates prone to seasonal infertility.

Future research should investigate these nano-based protocols’ scalability and field implementation across diverse management systems and environmental conditions. Long-term studies are necessary to evaluate the broader impacts on herd health, drug resistance, and economic sustainability. This research is essential to promote the adoption of these innovative nano-based treatments as a sustainable solution for improving fertility in dairy herds.

## Figures and Tables

**Figure 1 pharmaceutics-17-00274-f001:**
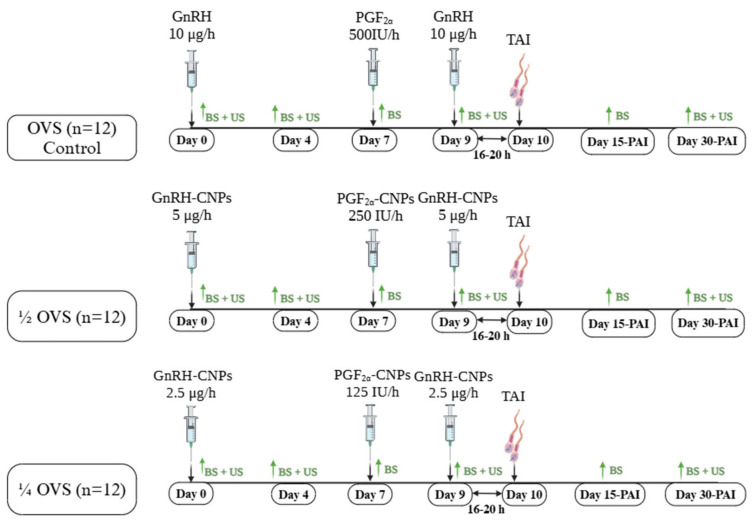
Schematic diagram of estrous synchronization protocols: Ovsynch (OVS) (control group) using bare GnRH (10 μg) and PGF_2α_ (500 μg) in Group 1; ½ OVS using GnRH-CNPs (5 μg) and PGF_2α_-CNPs (250 μg); and ¼ OVS using GnRH-CNPs (2.5 μg) and PGF_2α_-CNPs (125 μg). BS, blood sampling; US, ultrasonography; TAI, fixed-timed artificial insemination; PAI, post-artificial insemination. Created in BioRender.

**Figure 2 pharmaceutics-17-00274-f002:**
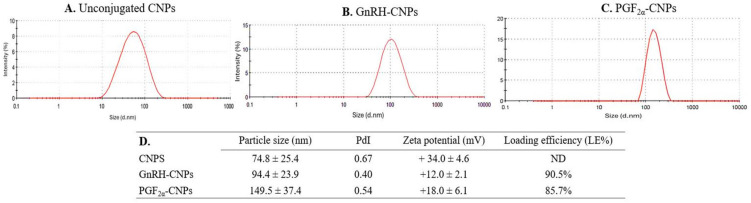
Illustrations show the particle size distribution of CNPs measured via Malvern Zetasizer before (**A**) and after (**B**,**C**) the hormonal conjugation, while (**D**) depicts the physicochemical features including average size (±SD), zeta potential (±SD), polydispersity index (PdI), and LE (%) of unconjugated CNPs, GnRH–CNPs, and PGF_2α_–CNPs.

**Figure 3 pharmaceutics-17-00274-f003:**
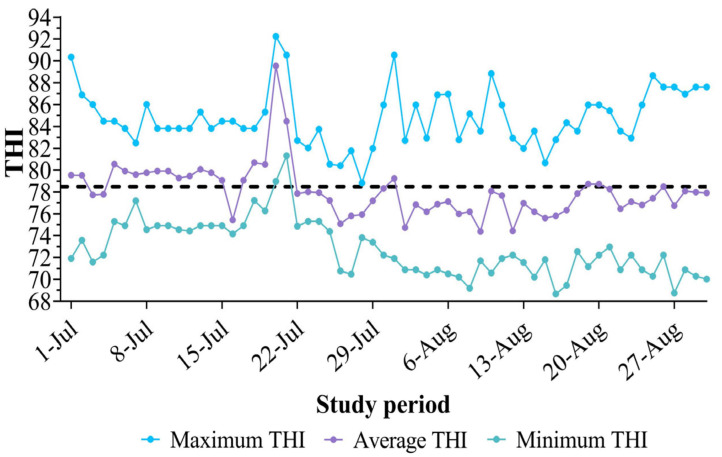
Daily maximum, minimum, and average THI values throughout the experimental period (1st of July to 31st of August). A THI value surpassing 72 indicates sever HS conditions for lactating cows.

**Figure 4 pharmaceutics-17-00274-f004:**
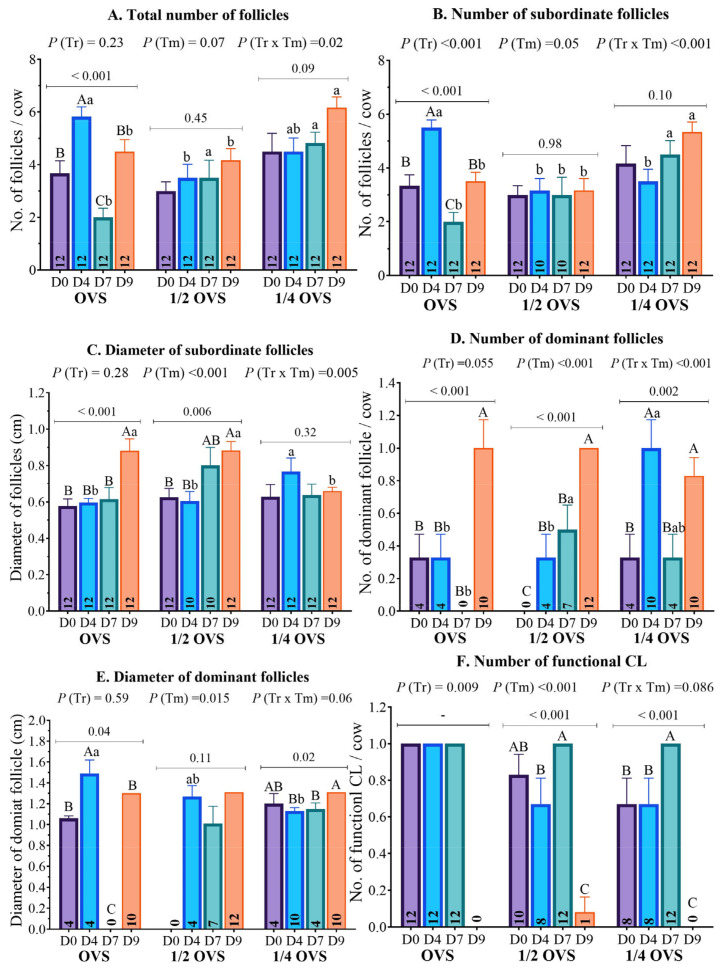
Effect of hormonal protocols (OVS, ½ OVS, and ¼ OVS) on average (**A**) total follicles number, (**B**) subordinate follicles number, (**C**) subordinate follicles diameter, (**D**) dominant follicles number, (**E**) dominant follicle diameter, and (**F**) functional CL number (>1 ng/mL) across different protocol days in heat-stressed dairy cows. Different lowercase and uppercase superscript letters refer to significant differences (*p* < 0.05) between and within treatments, respectively.

**Figure 5 pharmaceutics-17-00274-f005:**
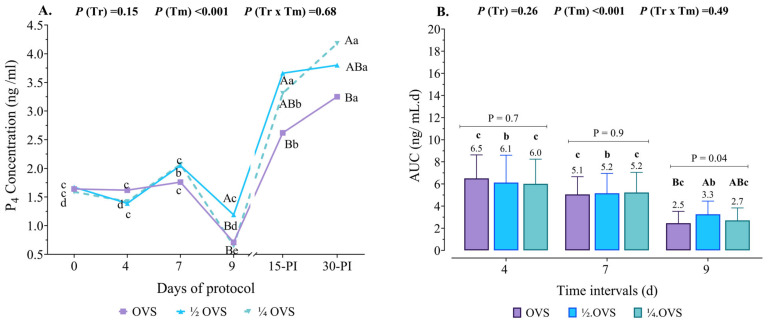
Panel (**A**) depicts the effect of different hormonal protocols (OVS, ½ OVS, and ¼ OVS) on serum P_4_ concentrations across different protocol days (0, 4, 7, 9, 15, 0 and 30 days post-AI). Panel (**B**) illustrates the changes in area under the curve (AUC) of P_4_ concentrations across distinct time intervals to monitor the immediate response of treatment in heat stressed dairy cows under different treatment protocols (OVS, ½ OVS, and ¼ OVS). The defined intervals were defined as follows: D4 (∆ AUC between days 0 and 4), D7 (∆ AUC between days 4 and 7), and D9 (∆ AUC between days 7 and 9). Different uppercase and lowercase superscript letters refer to significant differences (*p* < 0.05) between and within treatments, respectively.

**Table 1 pharmaceutics-17-00274-t001:** Effect of different hormonal protocols (OVS, ½ OVS, and ¼ OVS) on ovarian structures assessed by transrectal ultrasonography across different protocol days in heat-stressed dairy cows (mean ± SE).

Variables	Experimental Groups			*p*-Value
	OVS	½ OVS	¼ OVS	
Before the 1st GnRH administration (D0)				
Number of cows	12	12	12	
Total number of follicles	3.67 ± 0.48 (12)	3.00 ± 0.35 (12)	4.50 ± 0.69 (12)	0.15
Subordinate (≥0.3 to <1.0 cm) follicle number	3.33 ± 0.41 (12)	3.00 ± 0.35 (12)	4.17 ± 0.66 (12)	0.24
Subordinate (≥0.3 to <1.0 cm) follicle diameter	0.58 ± 0.04 (12)	0.63 ± 0.05 (12)	0.63 ± 0.07 (12)	0.76
Dominant (≥1.0 cm) follicle number	0.33 ± 0.14 (4)	0.00 ± 0.00 (0)	0.33 ± 0.14 (4)	0.07
Dominant (≥1.0 cm) follicle diameter	1.06 ± 0.02 (4)	- (0)	1.20 ± 0.10 (4)	0.20
Functional CL number	1.00 ± 0.00 (12)	0.83 ± 0.11 (10)	0.67 ± 0.14 (8)	0.09
Response to the 1st GnRH administration (D4)				
Total number of follicles	5.83 ± 0.37 ^a^ (12)	3.50 ± 0.52 ^b^ (12)	4.50 ± 0.52 ^ab^ (12)	0.005
Subordinate (≥0.3 to <1.0 cm) follicle number	5.50± 0.29 ^a^ (12)	3.17 ± 0.44 ^b^ (10)	3.50 ± 0.45 ^b^ (12)	<0.001
Subordinate (≥0.3 to <1.0 cm) follicle diameter	0.60 ± 0.02 ^b^ (12)	0.61 ± 0.05 ^b^ (10)	0.77 ± 0.07 ^a^ (12)	0.05
Dominant (≥1.0 cm) follicle number	0.33 ± 0.14 ^b^ (4)	0.33 ± 0.14 ^b^ (4)	1.00 ± 0.17 ^a^ (10)	0.005
Dominant (≥1.0 cm) follicle diameter	1.49 ± 0.13 ^a^ (4)	1.27 ± 0.10 ^ab^ (4)	1.09 ± 0.05 ^b^ (10)	0.008
Functional CL number	1.00 ± 0.00 (12)	0.67 ± 0.14 (8)	0.67 ± 0.14 (8)	0.08
Before PGF_2α_ administration (D7)				
Total number of follicles	2.00 ± 0.35 ^b^ (12)	3.50 ± 0.67 ^a^ (12)	4.83 ± 0.41 ^a^ (12)	0.001
Subordinate (≥0.3 to <1.0 cm) follicle number	2.00 ± 0.35 ^b^ (12)	3.00 ± 0.65 ^b^ (10)	4.50 ± 0.51 ^a^ (12)	0.007
Subordinate (≥0.3 to <1.0 cm) follicle diameter	0.62 ± 0.06 (12)	0.80 ± 0.10 (10)	0.64 ± 0.06 (12)	0.18
Dominant (≥1.0 cm) follicle number	0.00 ± 0.00 ^b^ (0)	0.50 ± 0.15 ^a^ (7)	0.33 ± 0.14 ^ab^ (4)	0.02
Dominant (≥1.0 cm) follicle diameter	- (0)	1.01 ± 0.17 (7)	1.13 ± 0.03 (4)	0.56
Functional CL number	1.00 ± 0.00 (12)	1.00 ± 0.00 (12)	1.00 ± 0.00 (12)	-
Day of the 2nd GnRH and response to PGF_2α_ administrations (D9)				
Total number of follicles	4.50 ± 0.45 ^b^ (12)	4.17 ± 0.44 ^b^ (12)	6.17 ± 0.41 ^a^ (12)	0.006
Subordinate (≥0.3 to <1.0 cm) follicle number	3.50 ± 0.34 ^b^ (12)	3.17 ± 0.44 ^b^ (12)	5.33 ± 0.38 ^a^ (12)	0.001
Subordinate (≥0.3 to <1.0 cm) follicle diameter	0.88 ± 0.07 ^a^ (12)	0.88 ± 0.05 ^a^ (12)	0.66 ± 0.02 ^b^ (12)	0.003
Dominant (≥1.0 cm) follicle number	1.01 ± 0.17 (10)	1.0 ± 0.00 (12)	0.83 ± 0.11 (10)	0.53
Dominant (≥1.0 cm) follicle diameter	1.30 ± 0.07 (10)	1.31 ± 0.05 (12)	1.15 ± 0.06 (10)	0.99
Functional CL number	0.00 ± 0.00 (0)	0.08 ± 0.29 (1)	0.00 ± 0.00 (0)	0.38

Different lowercase superscript letters within each row refer to significant differences (*p* < 0.05) between treatments. The functional classification indicates that the CL number must have a P_4_ concentration exceeding 1 ng/mL.

## Data Availability

Data is contained within the article.

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
