# Peer review of "Evaluating the Impact of Minimized GnRH and PGF2α Analogues-Loaded Chitosan Nanoparticles on Ovarian Activity and Fertility of Heat-Stressed Dairy Cows"

_pharmaceutics, 2025, doi:10.3390/pharmaceutics17020274_

Round 1
Reviewer 1 Report
Comments and Suggestions for Authors
The manuscript titled “Evaluating the Impact of Minimized GnRH and PGF2α Analogues-Loaded Chitosan Nanoparticles on Ovarian Activity and Fertility of Anestrous Dairy Cows” was done by Mohammed et al., there are some concerns need to clarify?
Major concerns:
1. The GnRH and PGF2α was always used to induce Ovsynch in cows, and the author only used Nano-drug delivery systems to deliver GnRH and PGF2α? Therefore, the advantage of Nano-drug delivery should be described in the introduction.
2. How to evaluate the disadvantage or detrimental to the cows?
3. The quality of Figures,especially Fig.1 should be improved.
4. How to evaluate the ovarian follicle or CL? Rectal examination or B-ultrasound?
Minor revision
1. In line 187, -20°C, please add a space between number and unit, please to check and revise all these problems in the entire manuscript.
2. In line 189, “the intra-assay and inter-assay coefficients of variation were 9.3% and 9.9%, respectively”, please to determine “93% and 99%”?
3. Please to check and revise the format of reference according to journal’s demand.
Author Response
Thank you for your comments. Please check the attachment.

Reviewer 2 Report
Comments and Suggestions for Authors
In this MS, Omar and colleagues evaluate the outcomes of reduced-dose Ovsynch protocols incorporating GnRH and PGF2α-loaded chitosan nanoparticles to enhance ovarian activity and fertility in anestrous dairy cows. The manuscript is clearly written and addresses an innovative approach to reproductive management in dairy cattle that bridges veterinary reproduction science and nanotechnology.
The study is well-designed, but the language needs some refinements. The paper establishes clear comparisons between standard and modified Ovsynch protocols. The sample size (3 groups of 12 cows) is modest but appropriate for preliminary findings. Moreover, follow-up studies on long-term outcomes (pregnancy confirmation at days 60-70) and validation in different environments would enhance the impact of this research.
Despite presenting innovative concepts, the manuscript needs stronger justification for using Ovsynch in anestrous cows, given the challenges associated with their lack of functional cyclicity. Ovsynch protocols are primarily designed to synchronize ovulation in cyclic cows; optimal results depend on a functional corpus luteum (CL) and an active estrous cycle. Therefore, the authors must present their (strong) justification for using Ovsynch in anestrous cows, given the challenges associated with their lack of functional cyclicity.
The Methods section covers nanoparticle synthesis, hormonal protocols, and statistical analyses. However, additional information is needed. In M&M, the authors must better characterize the ovarian activity of the females used in the study. When presenting the data respecting the presence of CL or the P4 measurements– figures 4 and 5 – we found that most cows in the control group had a CL at the beginning of treatment (in the treated groups, a smaller number of females – 8 to 10 cows - had a CL), which is in contradiction with their categorization as being anestrous. To clarify this issue, the authors should present a detailed description of the ovarian activity in each cow/group before (or at least on day 0 of the protocol). Otherwise, statements like those on lines 255 – 257 are incomprehensible and unmatch the figures' data.
Additional information is requested in the M&M: The reader is only informed that the cows were clinically healthy three months after calving and in their first to third lactation in an adequate BCS. Key variables influencing outcomes (e.g., nutrition, environmental conditions) are not fully discussed, which may affect reproducibility and interpretation of the results.
Still, in M&M, the authors should include information about the criteria set to assess the response to treatment. Otherwise, sentences like the one on line 276 are incomprehensible. What did the authors expect to be the response of a cow with a CL to the first GnRH administration? Or the prostaglandin injection on day 7? From graphs 4 and 5, I would suspect that most cows followed their natural cycles until prostaglandin administration, which induced luteolysis of the existing CL. Suppose cows started their treatment with an active CL. Could we claim that maintaining it after GnRH administration, followed by CL regression after induced luteolysis, is a failed outcome of the treatment? I do not think so!
Given the limited scale, the findings are significant but should be contextualized cautiously. The experimental methods align with the hypotheses, but the claim of high efficacy in anestrous cows might require validation with larger datasets. Even though the hypotheses are well-articulated and supported, broader validation is foreseen. There is limited discussion about how the findings apply broadly beyond the specific experimental conditions. Compare the results of the current study with others involving anestrous cows or nanoparticle-based hormone delivery systems, emphasizing similarities and differences. Discuss why these outcomes might differ in practical or field conditions. Validation in larger or diverse populations would strengthen these assertions. While the results suggest improved outcomes with reduced hormone doses, the claims about broad applicability need more support. In addition, the implications of nanotechnology-based treatments for practical field use (e.g., cost, accessibility, farmer acceptance) or some practical challenges (e.g., manufacturing nanoparticle-based hormones at scale) could have been explored. Although the discussion references the biological effects of GnRH and PGF2α nanoparticles, it does not deeply analyze the mechanisms behind their improved performance in anestrous cows or explore the potential mechanisms, such as enhanced bioavailability or sustained hormone release. I would like to challenge the authors to address this issue in the paper discussion.
Furthermore, the study's limitations are not adequately discussed; you should also explain how seasonal or environmental stressors might affect the generalizability of the findings. In the discussion, I also miss the assessment of putative weaknesses in the described experimental protocols, such as the variability in response among anestrous cows, the environmental side effects, etc.
The conclusions align with the presented data and emphasize the potential of nanoparticles to reduce hormone dosage while maintaining efficacy. Nonetheless, the authors should harden the conclusions by focusing on the study's preliminary nature and the need for further research to confirm findings across different herds and management systems.
Additional Recommendations:
- Proofread the text to ensure precision and eliminate typographical errors.
- Clarify if seasonal effects or other confounders influenced the results in anestrous cows.
- Discuss scalability and potential economic benefits of nanoparticle-based protocols.
- Expand on the practical implications of reduced hormone dosages
- By expanding the discussion section, the authors will be able to reduce the number of self-citations. Reliance on self-citations (up to 18% of the references used in the MS) might create a perception of bias.
A comment copy of the MS is attached with other minor suggestions or comments.

The language needs some correction in terms of grammar. Please proofread the text to ensure precision and eliminate typographical errors.
Author Response

(The authors gave the same response as above.)

Reviewer 3 Report
Comments and Suggestions for Authors
MS: pharmaceutics-3367395
Short Abstract: In the article entitled ‘Evaluation of the impact of chitosan nanoparticles loaded with minimised GnRH and PGF2α analogues on ovarian activity and fertility of anestrous dairy cows’, the authors evaluate in HF dairy cows (3 groups of 12 cows each) the efficacy of a nanoparticle-conjugated GnRH analogue (CNP) together with PGF2α CNP at half or quarter dose in a modified Ovsynch protocol on the reproductive performance of anestrous cows during the low breeding season in Egypt compared to the conventional dose of GnRH and PGF2α in the standard Ovsynch protocol. Indeed, the Ovsynch protocol is a widely used oestrus induction regime that significantly improves the reproductive efficiency of dairy cows. Recent applications in the production of nano-drugs have shown that nanoformulation of hormones can improve their biological activity by modifying their physicochemical properties. The authors observed that nanofabrication of GnRH and PGF2α reduced hormone doses by 50% and 25%, respectively, without compromising fertility outcomes in cows. Of the 3 protocols used, ¼.OVS showed a more robust follicular response and higher pregnancy rates than the conventional OVS protocol, indicating that GnRH-CNPs can improve ovarian activity and P4 concentrations during early pregnancy. These results support the use of GnRH-CNP and PGF2α-CNP as alternatives to conventional hormone treatments to improve reproductive efficiency and success in dairy cows, especially during the off-season.
I found the Manuscript interesting and in line with the aims of the Journal, but I have some suggestions for the authors, which I will list below, point by point:
Lines 23-47: Abstract. My suggestion is to rewrite it better and summarise it.
Lines 63-64: Elevated temperature leads to hyperthermia and heat stress, which impair fertility through various mechanisms [9,10].
This concept is fundamental and needs to be explained in detail.
Lines 156-161: I suggest adding details of diet and animal management, if available.
Lines 192-203: I suggest a correct description of the ultrasound device and probe in use (B-mode?).
lines 294-296: Notably, in our study, particle sizes increased after hormonal loading: unconjugated CNPs measured 74.8 ± 25.4 nm, while GnRH-CNPs and PGF2α-CNPs reached 94.4 ± 23.9 nm and 149.5 ± 37.4 nm, respectively.
Numerical data should be reported in the "Results" section and commented on in the "Discussion" section. Check, also in other parts of the "Discussion".
Lines 428-435: 5. Conclusions. My suggestion is to expand on the conclusions and, in particular, to highlight the practical implications of the study.
Author Response

(The authors gave the same response as above.)

Round 2
Reviewer 1 Report
Comments and Suggestions for Authors
The author has revised all my concerns, and the manuscript can be accepted.
Author Response
Responses to Academic Editor Notes:
The authors are grateful for the Reviewers' efforts in evaluating our manuscript. We appreciate your time spent reviewing. Before answering the most critical concerns, thank you for your valuable comments on the paper. We feel that the Reviewers’ comments and recommendations were reasonable, and we tried to consider them as much as possible while improving the manuscript. In our opinion, the reviewers' activities have contributed significantly to improving the quality of our paper. As you can see, we have made all the necessary corrections.
Reviewer 2.
In this MS, Omar and colleagues evaluate the outcomes of reduced-dose Ovsynch protocols incorporating GnRH and PGF2α-loaded chitosan nanoparticles to enhance ovarian activity and fertility in anestrous dairy cows. The manuscript is clearly written and addresses an innovative approach to reproductive management in dairy cattle that bridges veterinary reproduction science and nanotechnology.
Thank you for your constructive comments and suggestions. We have carefully considered each point and would like to respond as follows:
1.The study is well-designed, but the language needs some refinements. The paper establishes clear comparisons between standard and modified Ovsynch protocols. The sample size (3 groups of 12 cows) is modest but appropriate for preliminary findings. Moreover, follow-up studies on long-term outcomes (pregnancy confirmation at days 60-70) and validation in different environments would enhance the impact of this research.
Author: We appreciate your recognition of the study's design. We agree that our findings are preliminary, as there have been no previous studies discussing this point in dairy cows.
In response to your comment about long-term outcomes, we unfortunately do not have data on pregnancy confirmation at days 60-70 in this study. However, we acknowledge this limitation and have clearly included it in the discussion of research limitations (lines 489-500). We will definitely consider incorporating long-term follow-up assessments and validations across diverse environments in our future work to strengthen the implications and applicability of our findings.
Despite presenting innovative concepts, the manuscript needs stronger justification for using Ovsynch in anestrous cows, given the challenges associated with their lack of functional cyclicity. Ovsynch protocols are primarily designed to synchronize ovulation in cyclic cows; optimal results depend on a functional corpus luteum (CL) and an active estrous cycle. Therefore, the authors must present their (strong) justification for using Ovsynch in anestrous cows, given the challenges associated with their lack of functional cyclicity.
Author: Thank you for your insightful comment regarding the use of Ovsynch in anestrous cows. We understand the importance of clearly justifying our approach, particularly given the challenges associated with lack of functional cyclicity.
In our manuscript, we described the cows as anestrus, but upon review, we recognize that this terminology might have caused some confusion. These cows were categorized as anestrus because they did not exhibit visible estrous signs (silent heat) for more than 90 days postpartum during summer season. However, cows presented varying ovarian activities. To avoid ambiguity, we have decided to revise our terminology and clarify that while these cows did not show estrous signs, their ovarian status included different levels of activity (cyclic and non-acyclic cows). This adjustment will provide a more accurate description and strengthen the justification for applying Ovsynch protocols in this context.
Additionally, we have previously conducted research on buffaloes with true anestrus (smooth ovary syndrome), where the Ovsynch protocol, both in its conventional and nano-form, yielded positive outcomes. This prior research supports the use of Ovsynch in situations involving cows with impaired estrous cycle, further justifying its application in our study.
Gallab, R.S.; Hassanein, E.M.; Rashad, A.M.A.; El-, A.A. Maximizing the reproductive performances of anestrus dairy buffalo cows using GnRH analogue-loaded chitosan nanoparticles during the low breeding season. Anim. Reprod. Sci. 2022, 244, 107044, doi:10.1016/j.anireprosci.2022.107044.
The Methods section covers nanoparticle synthesis, hormonal protocols, and statistical analyses. However, additional information is needed. In M&M, the authors must better characterize the ovarian activity of the females used in the study. When presenting the data respecting the presence of CL or the P4 measurements– figures 4 and 5 – we found that most cows in the control group had a CL at the beginning of treatment (in the treated groups, a smaller number of females – 8 to 10 cows - had a CL), which is in contradiction with their categorization as being anestrous. To clarify this issue, the authors should present a detailed description of the ovarian activity in each cow/group before (or at least on day 0 of the protocol). Otherwise, statements like those on lines 255 – 257 are incomprehensible and unmatch the figures' data.
Author: Thank you for your valuable feedback. We appreciate your suggestion for further clarification regarding the ovarian activity of the animals in the study.
We would like to clarify that all animals in this study exhibited no signs of estrus, remaining in a state of silent estrus. Consequently, animals were randomly selected and distributed into groups without consideration of their ovarian activity. This factor may represent a significant challenge in our research and in interpreting the data, as the variations in ovarian activity among cows that did not display estrus resulted in differing responses to the established protocol.
We have revised the manuscript and included more details about the ovarian activity on day 0.
Additional information is requested in the M&M: The reader is only informed that the cows were clinically healthy three months after calving and in their first to third lactation in an adequate BCS. Key variables influencing outcomes (e.g., nutrition, environmental conditions) are not fully discussed, which may affect reproducibility and interpretation of the results.
Author: Thank you for your constructive comment. We appreciate your request for more detailed information in the M&M section.
We have already provided additional information about nutrition and environmental conditions that could potentially influence the outcomes and reproducibility of our results. Our focus was primarily on heat stress during the summer months, which is the main obstacle that prompted us to conduct this study to minimize its adverse effects on cows. So, we have included meteorological data for the calculation of the Temperature-Humidity Index (THI) to more accurately illustrate the severity of heat stress throughout the study period (lines 178-185). Also, we have included related interpretations and discussions of the results (lines 249-255, 342-355).
Still, in M&M, the authors should include information about the criteria set to assess the response to treatment. Otherwise, sentences like the one on line 276 are incomprehensible. What did the authors expect to be the response of a cow with a CL to the first GnRH administration? Or the prostaglandin injection on day 7? From graphs 4 and 5, I would suspect that most cows followed their natural cycles until prostaglandin administration, which induced luteolysis of the existing CL. Suppose cows started their treatment with an active CL. Could we claim that maintaining it after GnRH administration, followed by CL regression after induced luteolysis, is a failed outcome of the treatment? I do not think so!
Author: we have rephrased line 276 to provide clearer information regarding the treatment responses in the revised manuscript (lines 307-308).
Given the limited scale, the findings are significant but should be contextualized cautiously. The experimental methods align with the hypotheses, but the claim of high efficacy in anestrous cows might require validation with larger datasets. Even though the hypotheses are well-articulated and supported, broader validation is foreseen. There is limited discussion about how the findings apply broadly beyond the specific experimental conditions. Compare the results of the current study with others involving anestrous cows or nanoparticle-based hormone delivery systems, emphasizing similarities and differences. Discuss why these outcomes might differ in practical or field conditions. Validation in larger or diverse populations would strengthen these assertions. While the results suggest improved outcomes with reduced hormone doses, the claims about broad applicability need more support. In addition, the implications of nanotechnology-based treatments for practical field use (e.g., cost, accessibility, farmer acceptance) or some practical challenges (e.g., manufacturing nanoparticle-based hormones at scale) could have been explored. Although the discussion references the biological effects of GnRH and PGF2α nanoparticles, it does not deeply analyze the mechanisms behind their improved performance in anestrous cows or explore the potential mechanisms, such as enhanced bioavailability or sustained hormone release. I would like to challenge the authors to address this issue in the paper discussion.
Author: Thank you for your insightful feedback. We have mentioned all the aspects you raised by incorporating comparisons with other studies, discussing practical implications, and elaborating on the mechanisms behind the improved performance of GnRH and PGF2α nanoparticles. These additions are now included in the revised discussion and conclusion sections. We appreciate your suggestions, which have helped strengthen our manuscript.
Furthermore, the study's limitations are not adequately discussed; you should also explain how seasonal or environmental stressors might affect the generalizability of the findings. In the discussion, I also miss the assessment of putative weaknesses in the described experimental protocols, such as the variability in response among anestrous cows, the environmental side effects, etc.
Author: We have discussed the limitations and weaknesses in our experiment in the discussion section (lines 489-500).
The conclusions align with the presented data and emphasize the potential of nanoparticles to reduce hormone dosage while maintaining efficacy. Nonetheless, the authors should harden the conclusions by focusing on the study's preliminary nature and the need for further research to confirm findings across different herds and management systems.
Author: the conclusion is updated in the revised manuscript (lines 502-527)
Additional Recommendations:
Thank you for your constructive comments and suggestions. We have carefully considered each point in the revised manuscript and would like to respond as follows:
- Proofread the text to ensure precision and eliminate typographical errors.
Author: We have carefully proofread the manuscript to ensure precision and have corrected all typographical and formatting errors. I appreciate your thorough review and attention to detail.
- Clarify if seasonal effects or other confounders influenced the results in anestrous cows.
Author: Thank you for your constructive comment. We appreciate your request for more detailed information in the M&M section.
In our manuscript, we described the cows as anestrus, but upon review, we recognize that this terminology might have caused some confusion. These cows were categorized as anestrus because they did not exhibit visible estrous signs (silent heat) for more than 90 days postpartum during summer season. However, cows presented varying ovarian activities. To avoid ambiguity, we have decided to revise our terminology and clarify that while these cows did not show estrous signs, their ovarian status included different levels of activity (cyclic and non-acyclic cows). This adjustment will provide a more accurate description and strengthen the justification for applying Ovsynch protocols in this context.
We have already provided additional information about nutrition and environmental conditions that could potentially influence the outcomes and reproducibility of our results. Our focus was primarily on heat stress during the summer months, which is the main obstacle that prompted us to conduct this study to minimize its adverse effects on cows. So, we have included meteorological data for the calculation of the Temperature-Humidity Index (THI) to more accurately illustrate the severity of heat stress throughout the study period (lines 178-185). Also, we have included related interpretations and discussions of the results (lines 249-255, 342-355).
- Discuss scalability and potential economic benefits of nanoparticle-based protocols.
Author: Thank you for highlighting this important point. We have now expanded the discussion to include the potential scalability and economic benefits of nanoparticle-based protocols (lines 489-500).
- Expand on the practical implications of reduced hormone dosages
Author: Thank you for your valuable suggestion. We have expanded the discussion and conclusion to elaborate on the practical implications of reduced hormone dosages. (lines 489-527).
- By expanding the discussion section, the authors will be able to reduce the number of self-citations. Reliance on self-citations (up to 18% of the references used in the MS) might create a perception of bias.
Author: We have corrected it according to the request.
Reviewer 2 Report
Comments and Suggestions for Authors
The resubmitted MS was improved, but the authors need to include additional information to strengthen their findings and raise the overall interest in the research. The following concerns must be matched in full to support the quality of your findings.
1. The authors propose to present in the MS a cost-effect analysis of the use of the Chitosan-conjugated hormonal treatments in enhancing the reproductive performance of heat-stressed cows in Egypt. However, this analysis was not performed or presented in the MS. To fulfill this purpose, the authors will need to show a value for the unit of "costs" and characterize the unit of "gains" used in the analysis. Please clarify or amend the text according to the results presented herein
2. IN Results, the authors show the changes in different ovarian-related features (follicle size, CL numbers, progesterone levels) during treatment. However, they do not describe the ovarian activity at the beginning of the treatment, which is crucial to understanding the reported changes or the differences between the groups. To understand the descriptions of the number of follicles during the treatment, it is vital to know more about the ovarian activity of the cows in each group: Did they have CL on D0? How many presented CL on D0 in each group? The number and size of follicles surveyed during the treatment may be determined/affected by the day of the cycle in the cow at the beginning of the therapy. Therefore, this information is crucial for the study.
Looking at Figures 4 and 5, we can see that differences between the groups were reported on treatment days zero and 4. At that time, no luteolytic treatment was implemented, so the data respects the natural cycle of the cow. Under this assumption, it is mandatory to clarify:
Did the number of CLs change because of the normal dynamics of the stage of the ovarian cycle of cows? Or was it associated with the treatment? The same comments apply to the Progesterone measurements. The only logical explanation for the reported statistical differences on day zero and day 4 is the differences in the ovarian activity in cows from different groups. Therefore, the initial features of the cows' ovarian dynamics must be better characterized in the MS. Although the authors allude to it in the discussion, there is no support in the reported data, so the reader understands how that came up.
So, overall, it remains to be clarified in this MS the effect obtained with the OVS treatments. According to the manuscript's content, results may be biased by the moment of the cycle of each cow, which was not shown in the study.
3. The authors must provide additional information regarding the period when the experiment was developed and the moments of the day when the reported temperatures and humidity were observed. Were they collected at night? During the day? Do they represent an average for the days?
In some conditions, the heat stress of cows can be mitigated at night because the reduction in temperature and humidity allows the cows to freshen up and recover from the HTI effect during the day.
Other comments:
- in the abstract, please clarify whether the females were cyclic (presented ovarian activity despite not being observed in standing heat) or not at the OVS program's onset before reporting the results.
- When naming the groups, remove the dot between the fraction (1/4, 1/2) number and "OVS"
- Occasionally, some incongruencies were found between the text and the source referenced. Please revise the MS to avoid situations like those in lines 60-62 or 63-onward. See the commented copy attached for details
- correct across the MS: "estrus" when referring to the stage and "estrous" when referring to the cycle
- some suggestions in the grammar or sentence construction were included in the commented copy of your MS

Author Response
Reviewer 2
The resubmitted MS was improved, but the authors need to include additional information to strengthen their findings and raise the overall interest in the research. The following concerns must be matched in full to support the quality of your findings.
Authors: We appreciate your valuable and constructive feedback and suggestions to improve the quality of our manuscript.
- The authors propose to present in the MS a cost-effect analysis of the use of the Chitosan-conjugated hormonal treatments in enhancing the reproductive performance of heat-stressed cows in Egypt. However, this analysis was not performed or presented in the MS. To fulfill this purpose, the authors will need to show a value for the unit of "costs" and characterize the unit of "gains" used in the analysis. Please clarify or amend the text according to the results presented herein
Authors: We recognize the importance of conducting a coast effectiveness analysis to improve the implications of our findings. However, as this study represents a preliminary experiment, we did not include a detailed economic evaluation due to limited scope and available data. Therefore, we have modified our manuscript to focus only on the current results and have removed the coast effectiveness goal, which we plan to address in the upcoming work.
- In Results, the authors show the changes in different ovarian-related features (follicle size, CL numbers, progesterone levels) during treatment. However, they do not describe the ovarian activity at the beginning of the treatment, which is crucial to understanding the reported changes or the differences between the groups. To understand the descriptions of the number of follicles during the treatment, it is vital to know more about the ovarian activity of the cows in each group: Did they have CL on D0? How many presented CL on D0 in each group? The number and size of follicles surveyed during the treatment may be determined/affected by the day of the cycle in the cow at the beginning of the therapy. Therefore, this information is crucial for the study.
- Looking at Figures 4 and 5, we can see that differences between the groups were reported on treatment days zero and 4. At that time, no luteolytic treatment was implemented, so the data respects the natural cycle of the cow. Under this assumption, it is mandatory to clarify:
Did the number of CLs change because of the normal dynamics of the stage of the ovarian cycle of cows? Or was it associated with the treatment? The same comments apply to the Progesterone measurements. The only logical explanation for the reported statistical differences on day zero and day 4 is the differences in the ovarian activity in cows from different groups. Therefore, the initial features of the cows' ovarian dynamics must be better characterized in the MS. Although the authors allude to it in the discussion, there is no support in the reported data, so the reader understands how that came up.
So, overall, it remains to be clarified in this MS the effect obtained with the OVS treatments. According to the manuscript's content, results may be biased by the moment of the cycle of each cow, which was not shown in the study.
Authors: Thank you for your detailed comment. We have revised the result section to present the data more clearly and have added explanations to address your concerns. Specifically, we replaced the previous figures with a new table and figure to clarify more details regarding the ovarian activity at the beginning of the protocols (D0) and its improvement throughout the protocol days (Table 1 and Figure 4). Also, we have provided context for the observed changes based on treatments and the protocol days (Result section: lines 272 – 334, 369 - 377).
We also discuss whether the differences observed on D0 and D4 were due to the natural ovarian dynamics or treatment effects (Discussion section: lines 440 – 446, 475 – 478, and 507- 515)
- The authors must provide additional information regarding the period when the experiment was developed and the moments of the day when the reported temperatures and humidity were observed. Were they collected at night? During the day? Do they represent an average for the days?
Authors: Thank you for your insightful comments. We have taken the following actions to address these concerns and improve the manuscript:
- We have included additional data that provides a detailed THI condition during the experimental period. These data are based on hourly temperature and humidity readings recorded by a meteorological station located near the experiment site. This allowed us to calculate and present the minimum, maximum, and overall average THI values for the entire study period (July to August). These values were calculated for each day of the study, ensuring a clear representation of the environmental conditions.
- We have added additional explanations regarding the data collection methods, clarifying that the data were collected hourly, and we calculated the maximum, minimum, and average temperature and humidity to get maximum, minimum, and average THI during the study period (lines182-184)
- A new figure has been added to the manuscript (Figure 3) to visually illustrate the daily fluctuation of THI values during the experimental period.
- Results and discussion have been modified to provide a more comprehensive understanding of the experimental conditions (lines 251 – 259 and 426 – 427).
In some conditions, the heat stress of cows can be mitigated at night because the reduction in temperature and humidity allows the cows to freshen up and recover from the HTI effect during the day.
We would like to clarify that the experimental protocol started in early July, when the minimum, maximum, and average THI values consistently exceeded the threshold of heat stress of 72, indicating persistent heat stress, as shown in Figure 3. Although the minimum THI value started dropping below this threshold by mid-August, the early experimental period reflected sustained heat stress conditions throughout the day.
Other comments:
- in the abstract, please clarify whether the females were cyclic (presented ovarian activity despite not being observed in standing heat) or not at the OVS program's onset before reporting the results.
Authors: Done (lines 17 -18)
- When naming the groups, remove the dot between the fraction (1/4, 1/2) number and "OVS"
Authors: Done
- Occasionally, some incongruencies were found between the text and the source referenced. Please revise the MS to avoid situations like those in lines 60-62 or 63-onward. See the commented copy attached for details
Authors: Done
- correct across the MS: "estrus" when referring to the stage and "estrous" when referring to the cycle
Authors: Done
- some suggestions in the grammar or sentence construction were included in the commented copy of your MS
Authors: Done and modified as requested

Round 3
Reviewer 2 Report
Comments and Suggestions for Authors
The MS has generally improved across the revision process, but some questions that need clarification or a different approach during analysis remain:
#1. Regarding the reports on heat stress, the authors included a graph displaying the HTI variations throughout the study - it dramatically improves the study conditions' characterization. However, after analyzing the graph, I am not sure that the authors may conclude that cows were maintained in severe heat stress. Looking at the minimum HTI indexes, we would find that on most days, and even if we do not know the duration of the period represented the minimum values (I would assume it was the nights), it is possible that the cooling at night may contribute to relieving the heat stress felt during the day and allowing cows to recover from nefarious effects and adapt to environmental temperatures. I would suggest that you either remove the term "severe" and change the term "confirms" to "suggest". And to further discuss the topic in the discussion section
#2. To better study the changes in P4 concentrations during OVS treatment and compare them across the groups, the authors should use the area under the curve instead of the values for each blood collection point.
#3. Lines 482-485 - the sentence needs clarification because the information may conflict with data shown in the Results section (and it can be only because of the sentence grammar or absence of particularization)
Additional minor comments and/or suggestions were introduced in the commented copy of the MS attache to this report.

Author Response
Reviewer 2.
Thank you for your constructive comments and suggestions. We have carefully considered each point and would like to respond as follows:
The MS has generally improved across the revision process, but some questions that need clarification or a different approach during analysis remain:
#1. Regarding the reports on heat stress, the authors included a graph displaying the HTI variations throughout the study - it dramatically improves the study conditions' characterization. However, after analyzing the graph, I am not sure that the authors may conclude that cows were maintained in severe heat stress. Looking at the minimum HTI indexes, we would find that on most days, and even if we do not know the duration of the period represented the minimum values (I would assume it was the nights), it is possible that the cooling at night may contribute to relieving the heat stress felt during the day and allowing cows to recover from nefarious effects and adapt to environmental temperatures. I would suggest that you either remove the term "severe" and change the term "confirms" to "suggest". And to further discuss the topic in the discussion section
Author: Thank you for your suggestion to provide additional clarification. We have removed the term “sever” from MS, and we have replaced “confirm” with “suggest”, avoiding overinterpretation of the data. Additionally, we would like to clarify that the min. and max. temperature data represent 24-hour recorded data (not recorded at a specific time) obtained from a meteorological station. We extracted and calculated the min. and max. values using Excel. This information has been mentioned in the MS, but we appreciate the opportunity to reemphasize this point to ensure clarity.
#2. To better study the changes in P4 concentrations during OVS treatment and compare them across the groups, the authors should use the area under the curve instead of the values for each blood collection point.
Author: We recognize the importance of using the area under the curve (AUC) to demonstrate the cumulative changes in P4 levels over time. However, we also believe that the detailed P4 concentration values at each time point are essential for understanding the dynamics of P4 fluctuations throughout the protocol days.
So, to provide a comprehensive understanding of the data, we recommend retaining the original graph that displays P4 concentrations at individual time points (Figure 5A) and supplementing it with an additional graph illustrating the AUC (Figure 5B), as your recommendation. This dual representation will provide more understanding of data, highlighting the specific P4 levels at each sampling point and the overall hormonal trends throughout the treatment period in different experimental groups. We have included the graph of AUC in Figure 5B and included a detailed explanation of the calculation of AUC in M&M section (lines 217 -227) and updated the results to be consistent with the new graph (lines 405-423)
#3. Lines 482-485 - the sentence needs clarification because the information may conflict with data shown in the Results section (and it can be only because of the sentence grammar or absence of particularization)
Author: Thank you for pointing this out. We have revised the sentence to improve its clarity.
Additional minor comments and/or suggestions were introduced in the commented copy of the MS attache to this report.
Author: We have carefully reviewed and addressed all the modifications suggested in MS.
# Regarding the statement in line 370 “The functional classification indicates that the CL number must have a P4 concentration exceeding 1 ng/ml”, we believe it is essential to retain this statement as it clarifies that only the functional CLs were included in the table, based on measured P4 levels in our samples.

Round 4
Reviewer 2 Report
Comments and Suggestions for Authors
The revised MS responded to my concerns. In fact, respecting the use of the AUC to report differences in P4 levels during treatment, the authors went beyound the requested. I would recommend they remove the intervals from d9-fw, as it only make sense to retain it if discussing the results with the number of pregnant cows obtained with each treatment.
Otherwise, I recommend the publication of the MS with this minor change

Author Response
Reviewer 2
The revised MS responded to my concerns. In fact, respecting the use of the AUC to report differences in P4 levels during treatment, the authors went beyound the requested. I would recommend they remove the intervals from d9-fw, as it only make sense to retain it if discussing the results with the number of pregnant cows obtained with each treatment.
Otherwise, I recommend the publication of the MS with this minor change
Authors: Thank you for your constructive feedback, which has enhanced the quality of our MS.In accordance with your recommendation, we have removed the post-AI points from the AUC figure. Additionally, all suggested minor revisions have been implemented and highlighted in the revised MS.
